# Seroprevalence of anti-SARS-CoV-2 antibodies in Thai adults during the first three epidemic waves

Hatairat Lerdsamran[1], Anek Mungaomklang[2], Sopon Iamsirithaworn[3], Jarunee Prasertsopon[1], Witthawat Wiriyarat[4], Suthee Saritsiri[5], Ratikorn Anusorntanawat[6], Nirada Siriyakorn[7], Poj Intalapaporn[7], Somrak Sirikhetkon[2], Kantima Sangsiriwut[8], Worawat Dangsakul[9], Suteema Sawadpongpan[1], Nattakan Thinpan[1], Kuntida Kitidee[1], Pilailuk Okada[9], Ranida Techasuwanna[3], Noparat Mongkalangoon[3], Kriengkrai Prasert[10], Pilaipan Puthavathana[1]*

1 Center for Research and Innovation, Faculty of Medical Technology, Mahidol University, Nakhon Pathom, Thailand, 2 Institute for Urban Disease Control and Prevention, Department of Disease Control, Ministry of Public Health, Bangkok, Thailand, 3 Department of Disease Control, Ministry of Public Health, Nonthaburi, Thailand, 4 Faculty of Veterinary Science, Mahidol University, Nakhon Pathom, Thailand, 5 The 67th Public Health Center Thaweewatthana, Department of Health, Bangkok Metropolitan Administration, Bangkok, Thailand, 6 Chaophraya Yommarat Hospital, Office of the Permanent Secretary, Ministry of Public Health, Suphanburi, Thailand, 7 Rajavithi Hospital, Department of Medical Services, Ministry of Public Health, Bangkok, Thailand, 8 Department of Preventive and Social Medicine, Faculty of Medicine Siriraj Hospital, Mahidol University, Bangkok, Thailand, 9 Department of Medical Science, Ministry of Public Health, Nonthaburi, Thailand, 10 Nakhon Phanom Provincial Hospital, Department of Medical Services, Ministry of Public Health, Nakhon Phanom, Thailand

* pilaipan.put@mahidol.edu

**Data Availability Statement:** The minimal dataset is included in the paper and its Supporting Information files.

## Abstract

This study determined the presence of anti-SARS-CoV-2 antibodies in 4964 individuals, comprising 300 coronavirus disease-19 (COVID-19) prepandemic serum samples, 142 COVID-19 patients, 2113 individuals at risk due to their occupations, 1856 individuals at risk due to sharing workplaces or communities with COVID-19 patients, and 553 Thai citizens returning after spending extended periods of time in countries with a high disease prevalence. We recruited participants between May 2020 and May 2021, which spanned the first two epidemic waves and part of the third wave of the COVID-19 outbreaks in Thailand. Their sera were tested in a microneutralization and a chemiluminescence immunoassay for IgG against the N protein. Furthermore, we performed an immunofluorescence assay to resolve discordant results between the two assays. None of the prepandemic sera contained anti-SARS-CoV-2 antibodies, while antibodies developed in 88% (15 of 17) of the COVID-19 patients at 8–14 days and in 94–100% of the patients between 15 and 60 days after disease onset. Neutralizing antibodies persisted for at least 8 months, longer than IgG antibodies. Of the 2113 individuals at risk due to their occupation, none of the health providers, airport officers, or public transport drivers were seropositive, while antibodies were present in 0.44% of entertainment workers. Among the 1856 individuals at risk due to sharing workplaces or communities with COVID-19 patients, seropositivity was present in 1.9, 1.5, and 7.5% of the Bangkok residents during the three epidemic waves, respectively, and in 1.3% of the Chiang Mai people during the first epidemic wave. The antibody prevalence

**Funding:** This study was financially supported by the National Research Council of Thailand (NRCT), Grant No. NRCT 11/2563; the Department of Disease Control Ministry of Public Health, Thailand; and the National Science and Technology Development Agency (NSTDA), Thailand, Grant No. P-17-50551. The funders had no role in study design, data collection and analysis, decision to publish, or preparation of the manuscript.

**Competing interests:** The authors have declared that no competing interests exist.

varied between 6.5 and 47.0% in 553 Thai people returning from high-risk countries. This serosurveillance study found a low infection rate of SARS-CoV-2 in Thailand before the emergence of the Delta variant in late May 2021. The findings support the Ministry of Public Health's data, which are based on numbers of patients and contact tracing.

## Introduction

On 13 January 2020, Thailand was the first country to report a confirmed coronavirus disease-19 (COVID-19) case outside of China. The first indigenous case in Thailand occurred on 30 January 2020 in a local taxi driver who had no history of traveling abroad; an investigation suggested that he was exposed to severe acute respiratory syndrome coronavirus-2 (SARS-CoV-2) by a group of Chinese tourist passengers [1]. As of November 2021, Thailand had experienced 4 epidemic waves associated with different SARS-CoV-2 variants that emerged during the pandemic (Fig 1).

According to the Department of Disease Control (DDC), Ministry of Public Health (MOPH), the first epidemic wave began in January 2020, peaked in March–April, gradually declined, and ended on 14 December 2020. Large clusters occurring during this wave had linkages with attending boxing stadiums or entertainment venues in Bangkok. The strains causing this wave belonged to clades L, S, O, V, and G, likely reflecting multiple introductions of the virus into the country through tourism. The second wave was shorter, lasting from 15 December 2020 to 31 March 2021. This epidemic wave began among Myanmar migrant workers in a seafood market in Mahachai District, Samut Sakorn Province, located 37 km south of Bangkok. The causative virus belonged to the GH clade (G614 mutation), which was circulating in Myanmar during the second epidemic wave in August 2020 [2]. The third epidemic wave in Thailand began on 1 April 2021, again linked to nightlife entertainment venues, and might have been linked to the introduction of the Alpha variant (B.1.1.7) by the Thai people returning from Cambodia in the middle of March 2021 [3]. The Alpha variant has spread throughout

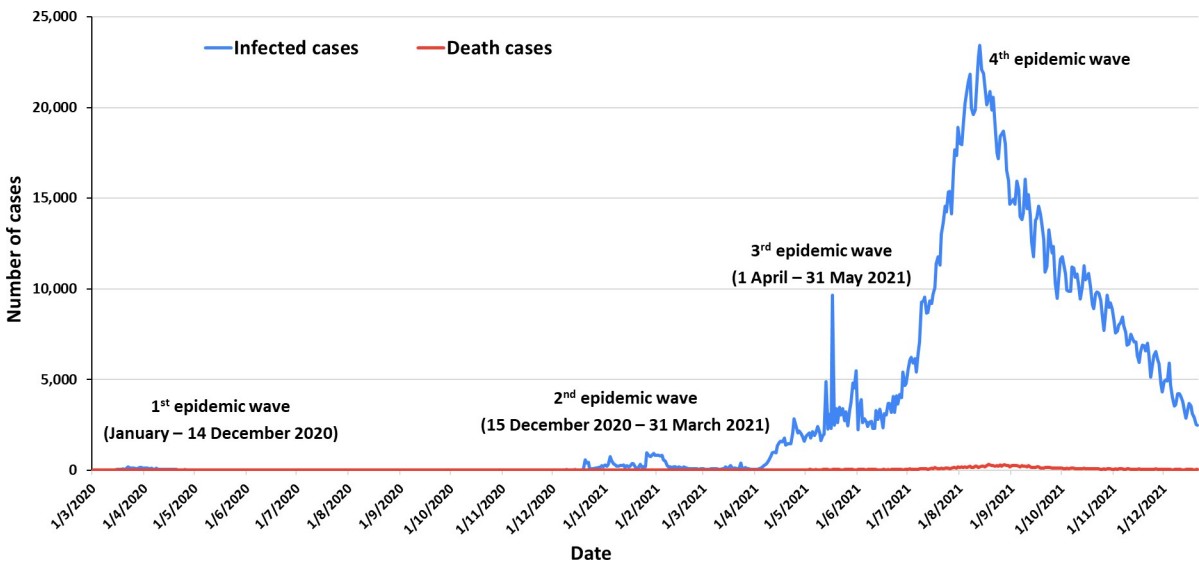

**Fig 1. SARS-CoV-2 infections in Thailand from 1 March 2020, through 21 December 2021.**

Cambodia, particularly in Phnom Penh, since February 2021 [4]. The Alpha variant spread to many parts of Thailand in the second week of April due to the movement of large numbers of people during the annual Songkran or water festival (the traditional Thai new year). Subsequently, the Delta variant, first detected on 7 May 2021 among workers in a large construction camp in Lak Si, Bangkok [5], spread countrywide, starting the fourth epidemic wave and causing a loss of control of the epidemic.

People infected with SARS-CoV-2 may develop antibodies targeting multiple viral proteins, regardless of whether they have symptoms. Therefore, serosurveillance is an important tool for estimating the magnitude of and monitoring the epidemic, especially since asymptomatic cases are common. A meta-analysis showed that as many as 35% of infected individuals were asymptomatic [6]. Serological data can also show the duration of antibody responses to COVID-19 infection, indicating partial protection from reinfection. Studies in Europe and the US have shown that neutralizing antibodies persist in a majority of COVID-19 patients for up to 13 months after infection [7], and patients with severe disease exhibit a higher level of neutralizing antibodies than those with milder disease [8]. The present study conducted a cross-sectional serosurveillance of anti-SARS-CoV-2 antibodies in Thai people in various risk groups in Bangkok and Chiang Mai Province from May 2020 to May 2021 (the duration spanned the first and second epidemic waves, and part of the third wave). The serological techniques used in this study included a microneutralization (microNT) assay, a chemiluminescence immunoassay (CLIA), and an indirect immunofluorescence assay (IFA).

## Materials and methods

### Ethical statement

The usage of human sera from participants older than 18 years was approved by the Mahidol University Central Institutional Review Board (MU-CIRB) under protocol number MU-COVID2020.001/2503, while the usage of prepandemic serum samples was approved by the same IRB under protocol number MU-CIRB 2018/014.1601.

### Study sites

Bangkok, the capital of Thailand, and Chiang Mai Province, located 696 km north of Bangkok, were chosen as the study sites due to their popularity as travel destinations, high population densities, and high numbers of SARS-CoV-2-infected patients.

### Study population

The study involved 4964 serum samples from 5 groups of participants, as follows. 1) Anonymized prepandemic COVID-19 serum samples collected from healthy adults between 2015 and 2019. 2) Anonymized archival serum samples from COVID-19 patients with no information of disease severity (sera were the leftover samples from clinical laboratory investigations). 3) Serum samples from people at risk due to their occupations (health personnel, airport officers, public transport drivers, and workers in entertainment venues (e.g., pubs, bars, and massage parlors)). 4) Serum samples from people at risk due to sharing workplaces or communities with COVID-19 patients. In the enrollment process for the participants included in Groups 2 and 3, epidemiologists explained the purpose of the study to obtain written consent for interviews about their demographics, occupation, workplace, residence, and general health condition, including a donation of 5–8 ml of blood, with specimens labeled using ID codes. 5) Serum samples from Thai citizens in state quarantines, who had arrived in Thailand after extended periods of work in countries with known SARS-CoV-2 outbreaks. Blood

**Table 1. Demographic data of participants in this study.**

| Group | Number of participants in each group | Number of participants with available data | Age, years | | Gender, n | | Blood collection date |
|---|---|---|---|---|---|---|---|
| | | | Range | Mean | Males | Females | |
| **Prepandemic serum samples** | | | | | | | |
| | 300 | 206 | 19–72 | 44.65 | 60 | 146 | 2015–2019 |
| **COVID-19 patients** | | | | | | | |
| | 142 | 113 | 19–89 | 39.42 | 58 | 55 | 24 February 2020–17 January 2021 |
| **Participants with at-risk occupations** | | | | | | | |
| Health personnel | 472 | 439 | 21–61 | 39.02 | 99 | 340 | 16 May 2020–17 May 2021 |
| Airport officers | 493 | 493 | 20–60 | 35.88 | 196 | 297 | 15 May 2020–2 December 2020 |
| Public transport drivers | 466 | 466 | 18–76 | 46.72 | 429 | 37 | 19 May 2020–6 January 2021 |
| Entertainment workers | 682 | 403 | 18–66 | 33.86 | 145 | 258 | 1 June 2020–2 December 2020 |
| **Participants who shared workplaces or lived in communities with reported COVID-19 cases** | | | | | | | |
| Bangkok | 1109 | 1066 | 18–85 | 40.97 | 488 | 578 | 14 May 2020–21 May 2021 |
| Chiang Mai | 747 | 747 | 18–90 | 56.38 | 163 | 584 | 3 June– 2 December 2020 |
| **Thai people returning from extended periods of work in high-risk countries** | | | | | | | |
| Qatar | 215 | 0 | NA | NA | NA | NA | 30 May– 17 June 2020 |
| Kuwait | 215 | 0 | NA | NA | NA | NA | 31 May– 17 June 2020 |
| Sudan | 77 | 77 | 25–52 | 35.47 | 76 | 1 | 12 October 2020 |
| Others | 46 | 42 | 22–63 | 41.43 | 18 | 24 | 17 June– 1 October 2020 |

NA = Not available

specimens were collected for anti-SARS-CoV-2 antibody testing [along with real-time reverse transcription-polymerase chain reaction (RT-PCR)] to support active case surveillance activities conducted by the Institute for Urban Disease Control and Prevention (IUDC), DDC, MoPH. For this group, an ethical review was waived under the authorization of the DDC, MOPH, as part of the emergency public health response to the pandemic. Nevertheless, the participants received explanations and gave verbal consent. The demographic data of these participants are shown in Table 1.

## Cell and virus cultivation

We cultivated Vero cells (African green monkey kidney cells—ATCC, CCL-81) in Eagle's minimum essential medium (EMEM) (Gibco, NY), supplemented with 10% fetal bovine serum (FBS) (Gibco, NY), penicillin, gentamycin, and amphotericin B. SARS-CoV-2 was isolated and propagated in Vero cells maintained in EMEM, supplemented with 2% FBS in a Biosafety Laboratory Level-3 facility. Using the Reed-Muench method, we calculated the virus concentration for a 50% tissue culture infective dose (TCID50) for further use in the microNT assay.

## Test viruses

This study used 3 SARS-CoV-2 isolates as the test viruses in the microNT assay. The purpose of this was to match the circulating virus at the time of the blood specimen collection. The SARS-CoV-2 isolate designated hCoV-19/Thailand/MUMT-4/2020, clade O (GISAID accession number EPI_ISL_493139) was used as the test virus for sera collected during the first epidemic wave, while hCoV-19/Thailand/MUMT-13/2021, clade GH (GISAID accession number

EPI_ISL_6267810), and hCoV-19/Thailand/MUMT-36/2021, clade GRY (B.1.1.7) (GISAID accession number EPI_ISL_6267895) were used as the test viruses for the second and third epidemic waves, respectively.

### Experimental design

In this cross-sectional surveillance study, we employed 3 serological techniques (microNT assay, CLIA-Architect IgG, and IFA) to detect anti-SARS-CoV-2 antibodies. The neutralizing (NT) antibodies were directed against the neutralizing epitopes present in the receptor-binding domain (RBD), N-terminal domain, and S2 domain in the spike protein [9–13]; Architect IgG was directed against the nucleoprotein (N) antigen, which is a more conserved protein [14, 15]. The microNT assay and Architect IgG were used to investigate every serum sample. When the results of the two assays were concordant, the test serum was considered either seropositive or seronegative. In the case of discordant results, the test serum was further investigated by IFA for total Ig against spike (S1) and N proteins to resolve the discordance. The microNT assay measured protective immunity, but the Architect IgG and IFA did not. This experimental design works well, as shown in our previous work [16]. The findings of this study are reported according to the Strengthening the Reporting of Observational Studies in Epidemiology (STROBE) statement guidelines as shown in S1 Table.

### Microneutralization assay

We used the cytopathic effect (CPE)-based microNT assay to determine the levels of NT antibodies against SARS-CoV-2 in the test sera. The assay employed Vero cell monolayers in 96-well microculture plates and SARS-CoV-2 at a final concentration of 100 TCID50/100 µl as the test virus. The methods followed those described in our previous studies [16, 17]. Briefly, the test serum was heat-inactivated and serially twofold diluted from 1:10 to 1:1280. A volume of 60 µl of each serum dilution was mixed with 60 µl of the test virus at a concentration of 200 TCID50/100 µl. After an hour of incubation at 37˚C, a volume of 100 µl of the virus-serum mixture was inoculated in duplicate into wells containing the Vero cell monolayer. The reaction plates were incubated at 37˚C for 3 days before the results were read. We defined the NT antibody titer as the highest reciprocal serum dilution that inhibited ≥50% CPE in the wells inoculated with the serum-virus mixture compared to the wells with the uninfected cell control. A titer of 10 or greater was considered positive for NT antibodies.

### Chemiluminescence immunoassay

CLIA using an Architect autoanalyzer (Abbott Laboratories, USA) is a two-step, fully automated immunoassay that qualitatively detects binding between the SARS-CoV-2 N antigen coated on paramagnetic microparticles and human IgG antibodies in the test sera. The assay required a minimum volume of 150 µl of test serum to fill the reaction cup. Acridinium-conjugated anti-human IgG bound to human IgG and then emitted chemiluminescence signals quantitated as relative light units (RLUs). It took approximately 1 hour to complete a test run. The level of SARS-CoV-2 IgG antibodies was directly correlated with the number of RLUs. An index value was established based on the ratio between the RLU of the kit positive sample (S) and the calibrator (C). A test serum with an S/C ratio ≥1.4 was considered positive for SARS-CoV-2 IgG antibodies.

### Indirect immunofluorescence assay

The IFA staining method has been previously described [16]. The assay employed SARS-CoV-2-infected Vero cells deposited on microscopic slides as the test antigens. To standardize the test antigen given the lot-to-lot variation, 50–75% of the infected cells must express N and S1 proteins when stained with specific monoclonal antibodies (Sino Biological, Beijing, China). Human serum at a dilution of 1:10 in phosphate-buffered saline was incubated with the infected cells for 60 minutes at 37°C, followed by the addition of polyclonal rabbit anti-human IgA, IgG, IgM, Kappa, and Lambda conjugated with fluorescein isothiocyanate (Dako, Glostrup, Denmark) in Evan's blue solution for 60 minutes. The stained viral antigens in the cytoplasm of the infected cells appeared apple green when examined under a fluorescence microscope.

### Statistical analysis

The R square ($R^2$), mean, and standard deviation (SD) were determined, and figures were drawn using GraphPad Prism version 8.4.3 for Windows (GraphPad Software, La Jolla, California, USA). The McNemar test was carried out, and the 95% confidence interval (95% CI) was calculated by SPSS Statistic software version 18.0.

## Results

### Determining anti-SARS-CoV-2 antibodies in prepandemic sera

Using CLIA, microNT, and IFA, 300 serum samples collected from healthy adults in the prepandemic period were examined for SARS-CoV-2 antibodies, if present. Two of the 300 (0.67%) participants were positive for IgG antibodies by CLIA, while all were negative by microNT assay. We further verified these two serum samples by IFA, and neither were positive. Our assay system could exclude the cross-reactivity between antibodies against previous human coronaviruses and SARS-CoV-2 antibodies.

### Prevalence of anti-SARS-CoV-2 antibodies in COVID-19 patients over time post symptoms

Using the microNT assay, CLIA, and IFA for confirmation, anti-SARS-CoV-2 antibodies were detected in 124 (87.3%) of the 142 COVID-19 patients (136 patients from the first epidemic wave, and 6 patients from the second wave). Eighty-eight percent (15 of 17) of the patients had antibody responses 8–14 days after the onset of symptoms. Positivity rose further to 94.1% (16 of 17) of the patients at 15–21 days, 100% (19 of 19) of the patients at 22–30 days, and 97% (32 of 33) of the patients at 31–60 days after the onset of symptoms (Fig 2). NT antibodies persisted for at least 8 months, as found in all 7 participants with a history of COVID-19, while IgG antibodies specific to the N protein were found in only 3 of these 7 participants. Nevertheless, the numbers of antibody-positive cases detected by the microNT assay and CLIA for IgG antibodies were not significantly different (McNemar test; $p = 0.65$). On the other hand, the NT antibody titers were not well correlated with the IgG levels ($R^2 = 0.6042$) (Fig 3).

### Prevalence of anti-SARS-CoV-2 antibodies in participants with selected occupations

The early COVID-19 outbreak in Thailand showed that individuals with some occupations had a higher risk of infection than individuals with other occupations. We conducted serosurveillance for anti-SARS-CoV-2 antibodies in 2113 participants with at-risk occupations (health

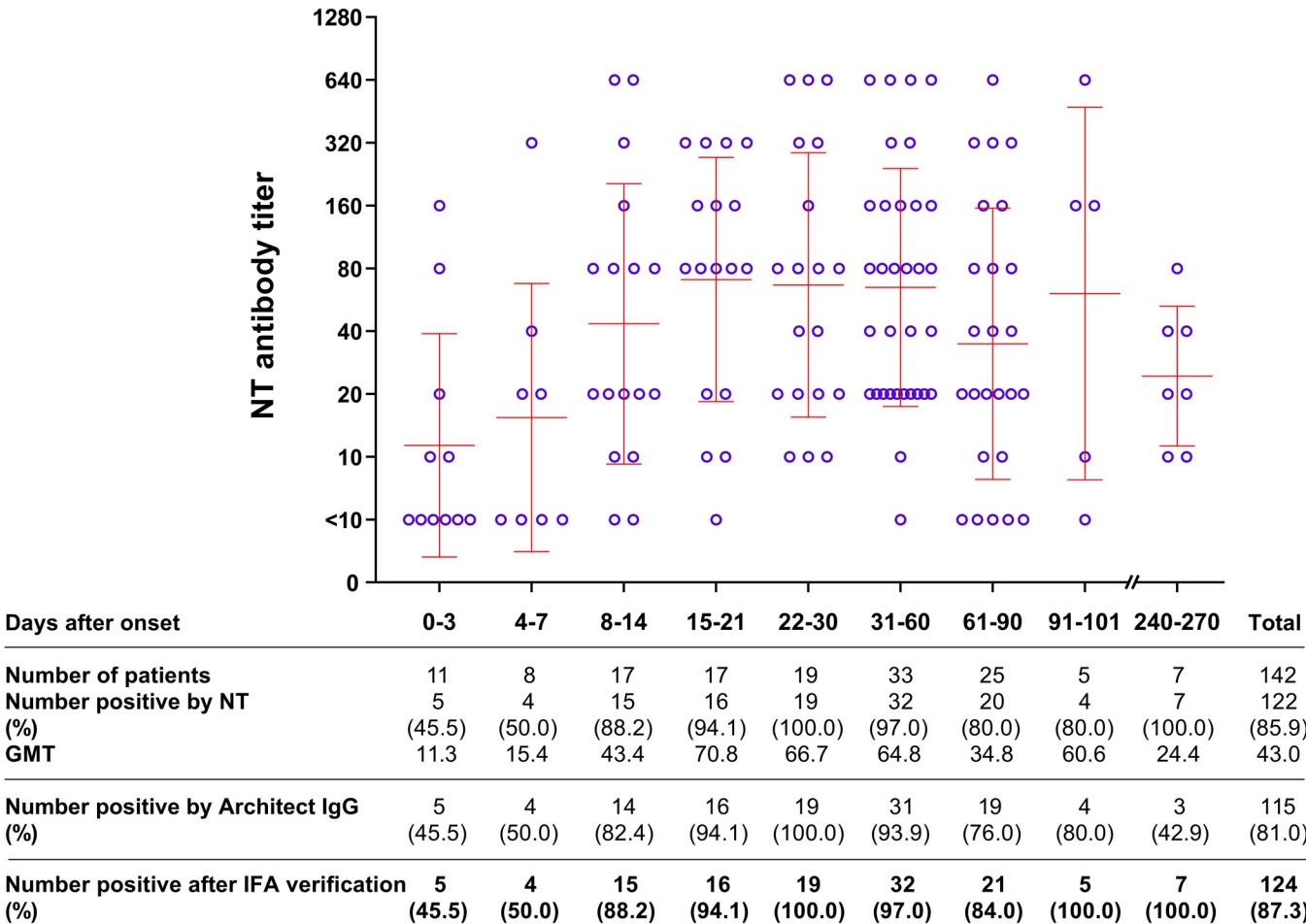

| Days after onset | 0-3 | 4-7 | 8-14 | 15-21 | 22-30 | 31-60 | 61-90 | 91-101 | 240-270 | Total |
|---|---|---|---|---|---|---|---|---|---|---|
| **Number of patients** | 11 | 8 | 17 | 17 | 19 | 33 | 25 | 5 | 7 | 142 |
| **Number positive by NT** | 5 | 4 | 15 | 16 | 19 | 32 | 20 | 4 | 7 | 122 |
| **(%)** | (45.5) | (50.0) | (88.2) | (94.1) | (100.0) | (97.0) | (80.0) | (80.0) | (100.0) | (85.9) |
| **GMT** | 11.3 | 15.4 | 43.4 | 70.8 | 66.7 | 64.8 | 34.8 | 60.6 | 24.4 | 43.0 |
| **Number positive by Architect IgG** | 5 | 4 | 14 | 16 | 19 | 31 | 19 | 4 | 3 | 115 |
| **(%)** | (45.5) | (50.0) | (82.4) | (94.1) | (100.0) | (93.9) | (76.0) | (80.0) | (42.9) | (81.0) |
| **Number positive after IFA verification** | 5 | 4 | 15 | 16 | 19 | 32 | 21 | 5 | 7 | 124 |
| **(%)** | (45.5) | (50.0) | (88.2) | (94.1) | (100.0) | (97.0) | (84.0) | (100.0) | (100.0) | (87.3) |

**Fig 2. Antibody responses in COVID-19 patients at different time points after the onset of disease symptoms.**

personnel, airport officers, public transport drivers, and workers in entertainment venues), as shown in Table 2. The serum samples from health personnel were collected during the three epidemic waves, while the others were collected from the first or second epidemic wave. The results showed that none of the 2113 participants had anti-SARS-CoV-2 antibodies, except for 3 (0.44%) of the 682 workers from entertainment venues.

## Prevalence of anti-SARS-CoV-2 antibodies in people at risk

We conducted serosurveillance for 1856 Thai people at risk due to sharing the same workplaces or living in the same communities as COVID-19 patients. The investigation showed that 1.9% (11 of 574), 1.5% (6 of 388), and 7.5% (11 of 147) of the people in Bangkok were seropositive for anti-SARS-CoV-2 antibodies during the three epidemic waves, respectively. In Chiang Mai, 1.3% (10 of 747) of the participants were seropositive during the first epidemic wave; 7 of them had a history of having COVID-19 during the prior 8 months (Table 3).

## Serological profiles of anti-SARS-CoV-2 antibodies in participants at risk

In Tables 2 and 3, we display the serological profiles of anti-SARS-CoV-2 antibodies in 3969 participants, grouped as 2113 people at risk due to their occupation (Table 2), and 1856 people

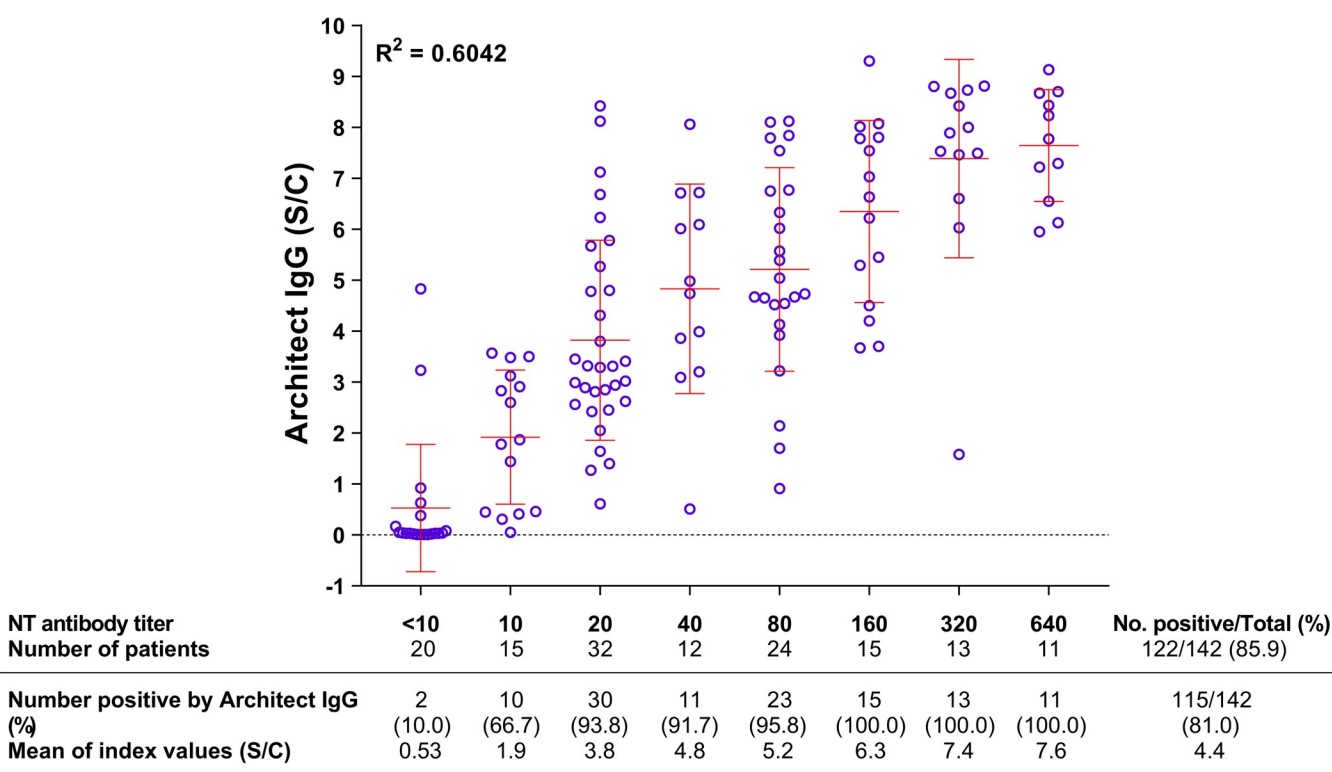

| NT antibody titer | <10 | 10 | 20 | 40 | 80 | 160 | 320 | 640 | No. positive/Total (%) |
|---|---|---|---|---|---|---|---|---|---|
| Number of patients | 20 | 15 | 32 | 12 | 24 | 15 | 13 | 11 | 122/142 (85.9) |
| Number positive by Architect IgG (%) | 2 (10.0) | 10 (66.7) | 30 (93.8) | 11 (91.7) | 23 (95.8) | 15 (100.0) | 13 (100.0) | 11 (100.0) | 115/142 (81.0) |
| Mean of index values (S/C) | 0.53 | 1.9 | 3.8 | 4.8 | 5.2 | 6.3 | 7.4 | 7.6 | 4.4 |

**Fig 3. Correlation between NT antibody titers and Architect IgG indices in COVID-19 patients.**

at risk due to sharing workplaces or communities with COVID-19 patients (Table 3). Using the microNT assay and Architect IgG, followed by IFA confirmation, 41 of the 3969 (1.0%) were seropositive for anti-SARS-CoV-2 antibodies. The number of seropositive cases detected by Architect IgG was slightly lower than that detected by microNT, but the difference was not significant (McNemar test; $p = 0.54$). Furthermore, the NT antibody titers were not well correlated with the IgG levels obtained by CLIA ($R^2 = 0.5908$) (Fig 4).

## Prevalence of anti-SARS-CoV-2 antibodies in travelers returning from high-risk areas

This study conducted serosurveillance for 553 Thai citizens returning after extended periods of work in countries with a high prevalence of SARS-CoV-2 infection between May and October 2020. Our results showed seroprevalences of 6.5–47.0% depending on the country (Table 4).

**Table 2. Seroprevalence of anti-SARS-CoV-2 antibodies in participants with at-risk occupations.**

| Group | Seroprevalence of anti-SARS-CoV-2 | | | |
|---|---|---|---|---|
| | (Number positive/Number tested) | | | |
| | 1st wave | 2nd wave | 3rd wave | Subtotal |
| Health personnel | 0/187 | 0/13 | 0/272 | 0/472 |
| Airport officers | 0/493 | - | - | 0/493 |
| Public transport drivers | 0/455 | 0/11 | - | 0/466 |
| Entertainment workers | 3/682 | - | - | 3/682 (0.44%) 95% CI = 0.08–1.29% |
| Total = 3/2113 (0.14%), 95% CI = 0.03–0.42% | | | | |

**Table 3. Seroprevalence of anti-SARS-CoV-2 antibodies in participants who shared workplaces or lived in communities with reported COVID-19 cases.**

| Province | Seroprevalence of anti-SARS-CoV-2 | | |
|---|---|---|---|
| | (Number positive/Number tested) | | |
| | **1st wave** | **2nd wave** | **3rd wave** |
| Bangkok | 11/574 (1.9%) 95% CI = 0.95–3.43% | 6/388 (1.5%) 95% CI = 0.56–3.37% | 11/147 (7.5%) 95% CI = 3.72–13.40% |
| Chiang Mai | 10*/747 (1.3%) 95% CI = 0.64–2.46% | - | - |
| **Subtotal** | 21/1321(1.6%) 95% CI = 0.98–2.43% | 6/388 (1.5) 95% CI = 0.56–3.37% | 11/147 (7.5%) 95% CI = 3.72–13.40% |
| | **Total = 38/1856 (2.04%), 95% CI = 1.44–2.81%** | | |

*Seven participants had a history of COVID-19 at 8 months previously.

## Characterization of seropositive sera

Of all 4964 serum samples investigated by microNT and CLIA with IFA verification, we obtained the final result of 320 seropositive samples. The seropositive results using individual assays is shown in S2 Table.

## Discussion

We employed 3 serological techniques to detect anti-SARS-CoV-2 antibodies in this cross-sectional serosurveillance study. The microNT assay and CLIA-Architect IgG were used to investigate every serum sample, and the IFA was used to verify only the sera with discordant results from the two methods. Our study showed that none of the prepandemic sera contained cross-reactive antibodies between previous human coronaviruses and SARS-CoV-2.

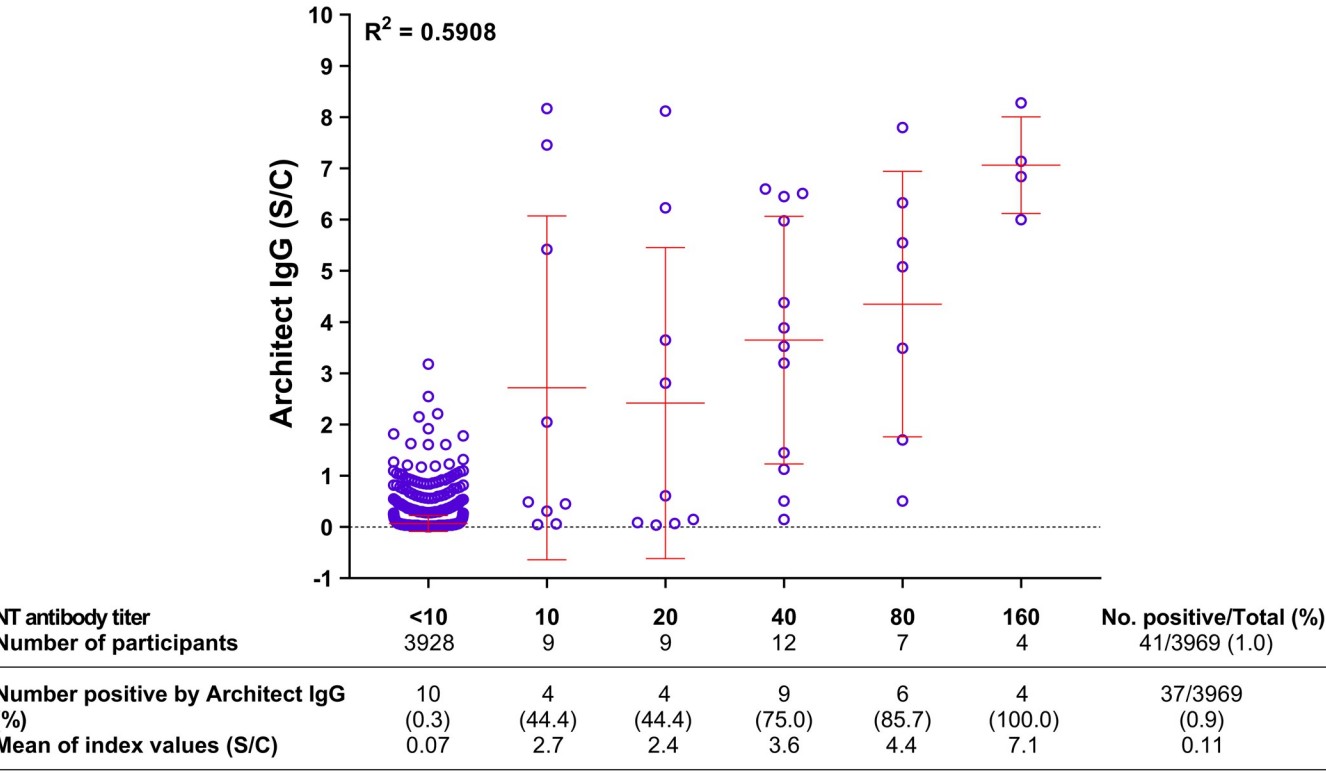

| NT antibody titer | <10 | 10 | 20 | 40 | 80 | 160 | No. positive/Total (%) |
|---|---|---|---|---|---|---|---|
| **Number of participants** | 3928 | 9 | 9 | 12 | 7 | 4 | 41/3969 (1.0) |
| **Number positive by Architect IgG (%)** | 10 (0.3) | 4 (44.4) | 4 (44.4) | 9 (75.0) | 6 (85.7) | 4 (100.0) | 37/3969 (0.9) |
| **Mean of index values (S/C)** | 0.07 | 2.7 | 2.4 | 3.6 | 4.4 | 7.1 | 0.11 |

**Fig 4. Correlation between NT antibody titers and Architect IgG indices in Thai people.**

**Table 4. Seroprevalence of anti-SARS-CoV-2 antibodies in Thai people returning from extended periods of work in high-risk countries.**

| Country | Date of blood collection | Number positive/Number tested (%) |
|---|---|---|
| Qatar | May–June 2020 | 14/215 (6.5%) 95% CI = 3.55–10.93% |
| Kuwait | May–June 2020 | 101/215 (47.0%) 95% CI = 38.27–57.08% |
| Sudan | October 2020 | 36/77 (46.8%) 95% CI = 32.75–64.72% |
| Others | June–October 2020 | 4/46 (8.7%) 95% CI = 2.28–22.36% |
| **Total = 155/553 (28.0%), 95% CI = 23.79–32.80%** | | |

Our study with COVID-19 patients showed that 15 of the 17 (88.2%) patients mounted a detectable antibody response 8–14 days after the onset of symptoms. However, the prevalence increased to as high as 94–100% in the subsequent 45 days. In addition, we found that anti-SARS-CoV-2 antibodies persisted for at least 8 months in all 7 individuals who had a history of COVID-19, while CLIA-Architect IgG antibodies to the N protein did not persist that long. This is similar to the findings of others who reported 8-month antibody persistence in patients with mild SARS-CoV-2 infections [18] and antibody persistence up to one year in one study [19]. Overall, the numbers of seropositive participants determined by the microNT assay and CLIA-Architect IgG were not significantly different. Nevertheless, the NT antibody titers were not well correlated with the CLIA-Architect IgG levels.

We investigated 4522 blood samples from multiple groups of participants between 14 May 2020, and 21 May 2021, spanning two epidemic waves and part of the third wave. Of the 2113 participants with at-risk occupations (472 health providers, 493 airport officers, 466 public transport drivers, and 682 workers in entertainment venues), only 0.14% (all 3 were from the last group) had detectable anti-SARS-CoV-2 antibodies. Furthermore, of the 1856 participants who shared workplaces or communities, only 38 (2.04%) were seropositive. Over time, the number of participants in Bangkok who had anti-SARS-CoV-2 antibodies was 1.9, 1.5, and increased to 7.5% during the 3 epidemic waves, respectively. None of the participants had received the COVID-19 vaccine at the time of blood collection, implying that all seropositive individuals in this study were naturally infected.

Due to low prevalence of SARS-CoV-2 infection in Thailand during the study period, our sample size may be too small to discover a few seropositive cases present in a certain population. Nevertheless, our low seroprevalence data was in line with those reported by the other Thai investigators. For example, the screening for anti-SARS-CoV-2 antibodies in 6,651 royal Thai army personnel between 1 July and 30 September 2020 yielded 41 seropositive participants as determined by Wondfo® rapid diagnosis. Nevertheless, only one participant could be confirmed by Euroimmune® ELISA, resulting in a seroprevalence of 0.015% [20]. Furthermore, the serosurveillance of 600 health care providers from 4 hospitals for one year after the first case was detected in Thailand yielded only one seropositive participant (seroprevalence 0.2%) [21]. Notably, this individual was positive for IgG antibodies against the S protein but not IgG antibodies against the N protein, as determined by Euroimmune® ELISA. These seroprevalence data strongly supported the low prevalence of infection reported by the MoPH, i.e., the cumulative numbers of 4237 cases with 60 deaths at the end of the first epidemic wave (from January to 14 December 2020), 28863 cases with 94 deaths at the end of the second epidemic wave (from 15 December 2020 to 31 March 2021), and 159792 cases with 1031 deaths (from 1 April to 31 May 2021).

Thailand received worldwide recognition for keeping the first epidemic wave well controlled. During the first epidemic wave, the cumulative case count per population was in the 10[th] percentile of countries worldwide [22]. As part of the control policy, the government

provided treatment and hospitalization, quarantine, and a laboratory diagnosis of SARS-CoV-2 infection at no cost. The prevalence of infection during the first wave peaked in March–April 2020, followed by a sharp decline. Less than 10 locally transmitted cases were detected daily from the middle of May through the end of the first epidemic wave. In total, there were 4237 cases with 60 deaths by the end of the first epidemic wave (14 December 2020). The high seroprevalence (6.5–47%) in Thai citizens returning from abroad suggested the seriousness of the outbreaks in those regions. Before the COVID-19 vaccine era, it was likely that the disease burden in Western countries was relatively more severe than that in most Asian countries. Several seroprevalence rates during the first epidemic wave varied from 0.16% in Tokyo [23] and 0.5% in Yamagata, Japan [24], 0.73% in India [25], 0.9% in Iceland [26], 3.8% in Israel [27], 5% in Spain (range <0.3 to >10% according to the locations) [28], to as high as 10.8% in Geneva [29], and approximately 20% in New York City [30].

The management of nonpharmaceutical interventions in Thailand was efficient. No people protested against wearing masks in public. They followed the suggestions on soft lockdowns, social distancing, working from home, and personal hygiene. A vital benefit came from the assistance of approximately one million village health volunteers who are part of the public health system and have worked nationwide since the time of H5N1 avian influenza. These volunteers assist with health education, active case finding, and communication between health authorities and communities. For example, each volunteer is assigned to take care of approximately 10 houses. Nevertheless, the occurrence of the second and third epidemic waves occurred very abruptly from the introduction of the newer variants, the GH clade and the Alpha variant, respectively. We cannot deny that these outbreaks were due to illegal activities, including the cross-border movement of migrant workers and people who gamble [31–33].

Before the third epidemic wave trended down, the outbreak of SARS-CoV-2 worsened due to the introduction of the Delta variant (which is more transmissible and virulent), which began the fourth epidemic wave. The infection rate peaked in August 2021, when more than 20000 cases were reported daily for weeks. Nevertheless, the Department of Medical Science, MoPH reports that the newer variant, Omicron, which was introduced into Thailand in November 2021, has spread rapidly and almost completely replaced the Delta variant at present.

Nonpharmaceutical intervention and vaccination should be practiced to slow viral infection and disease development. The MoPH first launched COVID-19 vaccinations for health providers and selected groups of people in late February 2021. By 16 March 2022, approximately 127 M doses were administered, which accounted for 71.6% of the population who received complete vaccination with 2 doses [34]. Even though vaccines prevent hospitalization, severe disease, and death, they do not prevent infection. As of 18 March 2022, there have been 3,303,169 infected patients with 24,075 deaths since the pandemic began in Thailand [34].

At present, seroprevalence studies will encounter difficulties in differentiating between natural infection and vaccination. Nevertheless, none of our participants received the COVID-19 vaccine, implying that all seropositive patients in this study were naturally infected. Therefore, our seroprevalence data provide a document for the low prevalence of SARS-CoV-2 infection in Thailand during the first year of the pandemic.

## Supporting information

**S1 Table. Strengthening the Reporting of Observational Studies in Epidemiology (STROBE) checklist.**
(DOCX)

**S2 Table. Verification of SARS-CoV-2 seropositive cases using microNT, CLIA-Architect IgG, and IFA.**
(DOCX)

## Acknowledgments

We thank Abbott Laboratories for providing the chemiluminescence test kits. We also thank Chinda Kanoksinsombat for coordinating the study, and Tipsuda Chanmanee, Rumporn Kularb, and Charlearnsak Thapyotha for their laboratory assistance.

## Author Contributions

**Conceptualization:** Witthawat Wiriyarat, Pilaipan Puthavathana.

**Data curation:** Anek Mungaomklang, Sopon Iamsirithaworn, Suthee Saritsiri, Ratikorn Anusorntanawat, Nirada Siriyakorn, Somrak Sirikhetkon, Worawat Dangsakul, Suteema Sawadpongpan, Nattakan Thinpan, Ranida Techasuwanna, Noparat Mongkalangoon, Kriengkrai Prasert.

**Formal analysis:** Hatairat Lerdsamran, Jarunee Prasertsopon, Kantima Sangsiriwut, Suteema Sawadpongpan.

**Funding acquisition:** Pilaipan Puthavathana.

**Investigation:** Anek Mungaomklang, Suthee Saritsiri, Poj Intalapaporn.

**Methodology:** Hatairat Lerdsamran, Jarunee Prasertsopon, Kantima Sangsiriwut, Nattakan Thinpan.

**Project administration:** Pilaipan Puthavathana.

**Resources:** Anek Mungaomklang, Sopon Iamsirithaworn, Witthawat Wiriyarat, Suthee Saritsiri, Ratikorn Anusorntanawat, Nirada Siriyakorn, Poj Intalapaporn, Somrak Sirikhetkon, Worawat Dangsakul, Kuntida Kitidee, Pilailuk Okada, Ranida Techasuwanna, Noparat Mongkalangoon, Kriengkrai Prasert.

**Software:** Kantima Sangsiriwut, Suteema Sawadpongpan.

**Supervision:** Pilaipan Puthavathana.

**Visualization:** Suteema Sawadpongpan.

**Writing – original draft:** Hatairat Lerdsamran.

**Writing – review & editing:** Pilaipan Puthavathana.

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
