## [Decision Letter · Decision Letter 0]

7 Feb 2022

PONE-D-22-01456Seroprevalence of anti-SARS coronavirus 2 antibodies in Thai adults during the first three epidemic wavesPLOS ONE

Dear Dr. Puthavathana,

Thank you for submitting your manuscript to PLOS ONE. After careful consideration, we feel that it has merit but does not fully meet PLOS ONE’s publication criteria as it currently stands. Therefore, we invite you to submit a revised version of the manuscript that addresses the points raised during the review process.

We look forward to receiving your revised manuscript.

Kind regards,

Jayanta Bhattacharya

Academic Editor

PLOS ONE

Journal Requirements:

Reviewers' comments:

Reviewer's Responses to Questions

**Comments to the Author**

1. Is the manuscript technically sound, and do the data support the conclusions?

Reviewer #1: Yes

Reviewer #2: No

Reviewer #3: Yes

2. Has the statistical analysis been performed appropriately and rigorously? 

Reviewer #1: Yes

Reviewer #2: Yes

Reviewer #3: Yes

3. Have the authors made all data underlying the findings in their manuscript fully available?

Reviewer #1: Yes

Reviewer #2: No

Reviewer #3: Yes

4. Is the manuscript presented in an intelligible fashion and written in standard English?

Reviewer #1: Yes

Reviewer #2: Yes

Reviewer #3: No

5. Review Comments to the Author

Reviewer #1: The manuscript is straightforward wherein the authors have performed a sero survey in different subsets of individuals in Thailand. Based on their binding and neutralization assays they have calculated percent positivity in the population. Of these, they also have 142 COVID-19 patients.

The following points require clarification and revision:

1. There is no handle on when immunization was introduced in Thailand. The authors must state this somewhere in their manuscript such that the reader can understand that all the responses being studied are in context with natural infection - this is important especially in context of health care workers studied since this group was typically the one that countries prioritized to immunize.

2. What is the baseline for all the assays? The authors must revise their data based on running some pre-pandemic healthy individuals and include the data points in the graphs presented. THis will allow the reader a better understanding of what is positive versus negative. Otherwise, there is a lot of heterogeneity in the analysis.

3. THe 142 patients that were sampled are shown stratified along different days post onset of clinical symptoms. Were any sequential bleeds collected? If so, it would be more fruitful to analyze the data in a kinetic manner to understand the level of antibody drop within the same individual.

4. Why was IgM analysis not performed in the patient samples? The data looks skewed with a peak because IgM responses were not studied - is it possible that all the early samples post onset of clinical symptoms would show higher antibody titers if IgM was analyzed? After all it is a primary covid-19 infection and one would expect a good IgM antibody response early.

5. There are 2 or 3 sero prevalence studies from Thailand. The authors have not acknowledged or discussed them. It is important to understand what information are these studies collectively providing.

6. The authors also do not discuss what this low seropositivity means in the coming years? Some level of discussion based on studies and findings from other parts of the world with respect to vaccination strategies, ability of a mostly naive population to handle variants etc is required to shed light on the importance of their study in the current time.

Reviewer #2: At the outset, I would like to appreciate the authors for conducting this study. Sero-epidemiology studies done at local levels inform the public health authorities to tailor their response to the context. In this study, the authors have attempted to determine the anti-SARS-CoV-2 antibody status of 4111 Thai people from May 2020 to April 2021, a period which spanned the first two and part of the third epidemic wave of the COVID-19 in Thailand. While this is an important study, I have specific concerns on the methodology and technical aspects of the study such as the choice and methods of antibody assays.

Major concerns:

1. Since this study is reported as a seroprevalence study, the sampling technique becomes the most critical aspect for interpretation of results. Seroprevalence studies are aimed to estimate the prevalence of the studies in the community and specific target populations. In order to achieve that the sampling strategy should be unbiased and representative and sample size should be adequate for the estimate to be precise. Without the information on these, the results of this study are difficult to interpret.

2. The seroprevalence could have been due to a natural infection or vaccination. The results have to be stratified by these to clearly understand the burden of infections during the different surges.

Minor concerns:

1. The demographic characteristics of the participants have to be provided as table-1.

2. It would be advisable to provide a STROBE checklist as supplementary.

3. A break-up of seroprevalence by the individual assays (in supplementary) will be informative to understand the discordance between the assays.

Reviewer #3: Reviewer’s comments on manuscript by Puthavathana and colleagues

Seroprevalence of anti-SARS coronavirus 2 antibodies in Thai adults during the first

three epidemic waves

This manuscript investigated the SARS-CoV-2 sero-prevalence in over 4,000 Thai individuals in the Bangkok and Chiang Mai regions. Various groups, including hospitalized individuals and those with high risk for exposure to infection, were sampled over the course of three epidemic waves and prior to the loss of SARS-CoV-2 containment by non-pharmaceutical interventions. Overall, the study was well executed given than three measures of sero-prevalence were used, and the manuscript is clear, but requires some editing with respect to grammatical errors.

Minor comments:

Figure 1 – reduce the breakdown for number of days post-onset to more biologically relevant intervals. For example, 0-10 days can be group as in general, antibodies against SARS-CoV-2 are detectable 10 days post symptom onset.

I would move the description of how the results of the three serological tests were used (line 255-262) to the start of the results section.

Line 303 – please provide a reference for the statement: “We cannot deny that these outbreaks due

6. PLOS authors have the option to publish the peer review history of their article (what does this mean?). If published, this will include your full peer review and any attached files.

Reviewer #1: No

Reviewer #2: **Yes: **Ramachandran Thiruvengadam

Reviewer #3: No

---

## [Author Response · Author response to Decision Letter 0]

25 Mar 2022

Reviewer #1: The manuscript is straightforward wherein the authors have performed a sero survey in different subsets of individuals in Thailand. Based on their binding and neutralization assays they have calculated percent positivity in the population. Of these, they also have 142 COVID-19 patients.

The following points require clarification and revision:

1. There is no handle on when immunization was introduced in Thailand. The authors must state this somewhere in their manuscript such that the reader can understand that all the responses being studied are in context with natural infection - this is important especially in context of health care workers studied since this group was typically the one that countries prioritized to immunize.

Response: In the Discussion of the submitted manuscript, we have already stated that “The MoPH first launched COVID-19 vaccinations for health providers and selected groups of people in late February 2021” (Please see line number 360). To make it more straightforward, we add one more sentence in the revised manuscript “None of the participants had received the COVID-19 vaccine at the time of blood collection, implying that all seropositive individuals in this study were naturally infected.” (Please see line number 308)

2. What is the baseline for all the assays? The authors must revise their data based on running some pre-pandemic healthy individuals and include the data points in the graphs presented. This will allow the reader a better understanding of what is positive versus negative. Otherwise, there is a lot of heterogeneity in the analysis.

Response: As shown below, we add one more paragraph on determining the baseline data of anti-SARS-CoV-2 antibodies in prepandemic sera in the Result (Line numbers 205-211). Accordingly, we have to add more information in the paragraph on “Study population” in Materials and methods (Line numbers 112-114). 

Lines 205-211:

Determining anti-SARS-CoV-2 antibodies in prepandemic sera

Using CLIA, microNT, and IFA, 300 serum samples collected from healthy adults in the prepandemic period were examined for SARS-CoV-2 antibodies, if present. Two of the 300 (0.67%) participants were positive for IgG antibodies by CLIA, while all were negative by microNT assay. We further verified these two serum samples by IFA, and neither were positive. Our assay system could exclude the cross-reactivity between antibodies against previous human coronaviruses and SARS-CoV-2 antibodies.

Lines 112-114 in Study population:

The study involved 4964 serum samples from 5 groups of participants, as follows. 1) Anonymized prepandemic COVID-19 serum samples collected from healthy adults between 2015 and 2019. 

3. The 142 patients that were sampled are shown stratified along different days post onset of clinical symptoms. Were any sequential bleeds collected? If so, it would be more fruitful to analyze the data in a kinetic manner to understand the level of antibody drop within the same individual.

Response: We agree with this comment. Unfortunately, we could not obtain any sequential samples as these patients’sera were the leftover after clinical laboratory investigations. 

4. Why was IgM analysis not performed in the patient samples? The data looks skewed with a peak because IgM responses were not studied - is it possible that all the early samples post onset of clinical symptoms would show higher antibody titers if IgM was analyzed? After all it is a primary covid-19 infection and one would expect a good IgM antibody response early.

Response: Most of patients in this study developed COVID-19 during the first epidemic wave of Thailand, which peaked in March 2020. Regarding antibody testing, only IgG but not IgM assay has just been available in late 2020. However, it was not until early 2021 that IgM assay was available in Thailand. As only a limited volume of archival sera, several samples we used up several samples on verification of many antibody testing methods, and most of the remaining samples were not adequate for IgM assay. 

5. There are 2 or 3 seroprevalence studies from Thailand. The authors have not acknowledged or discussed them. It is important to understand what information are these studies collectively providing.

Response: Thank you for your comment. In the revised manuscript, we discussed two articles (reference numbers 20 and 21) from the other Thai investigators. Please see line numbers 311-321 as shown below. 

Lines 311-321:

Due to low prevalence of SARS-CoV-2 infection in Thailand during the study period, our sample size may be too small to discover a few seropositive cases present in a certain population. Nevertheless, our low seroprevalence data was in line with those reported by the other Thai investigators. For example, the screening for anti-SARS-CoV-2 antibodies in 6,651 royal Thai army personnel between 1 July and 30 September 2020 yielded 41 seropositive participants as determined by Wondfo® rapid diagnosis. Nevertheless, only one participant could be confirmed by Euroimmune® ELISA, resulting in a seroprevalence of 0.015% [20]. Furthermore, the serosurveillance of 600 health care providers from 4 hospitals for one year after the first case was detected in Thailand yielded only one seropositive participant (seroprevalence 0.2%) [21]. Notably, this individual was positive for IgG antibodies against the S protein but not IgG antibodies against the N protein, as determined by Euroimmune® ELISA...

6. The authors also do not discuss what this low seropositivity means in the coming years? Some level of discussion based on studies and findings from other parts of the world with respect to vaccination strategies, ability of a mostly naive population to handle variants etc is required to shed light on the importance of their study in the current time.

Response: As shown below, we add more seroprevalence data in other countries in the Discussion. Please see line numbers 336-341.

Lines 336-341:

Before the COVID-19 vaccine era, it was likely that the disease burden in Western countries was relatively more severe than that in most Asian countries. Several seroprevalence rates during the first epidemic wave varied from 0.16% in Tokyo [23] and 0.5% in Yamagata, Japan [24], 0.73% in India [25], 0.9% in Iceland [26], 3.8% in Israel [27], 5% in Spain (range <0.3 to >10% according to the locations) [28], to as high as 10.8% in Geneva [29], and approximately 20% in New York City [30].

Reviewer #2: At the outset, I would like to appreciate the authors for conducting this study. Sero-epidemiology studies done at local levels inform the public health authorities to tailor their response to the context. In this study, the authors have attempted to determine the anti-SARS-CoV-2 antibody status of 4111 Thai people from May 2020 to April 2021, a period which spanned the first two and part of the third epidemic wave of the COVID-19 in Thailand. While this is an important study, I have specific concerns on the methodology and technical aspects of the study such as the choice and methods of antibody assays.

Major concerns:

1. Since this study is reported as a seroprevalence study, the sampling technique becomes the most critical aspect for interpretation of results. Seroprevalence studies are aimed to estimate the prevalence of the studies in the community and specific target populations. In order to achieve that the sampling strategy should be unbiased and representative and sample size should be adequate for the estimate to be precise. Without the information on these, the results of this study are difficult to interpret.

Response: Due to the low number of reported COVID-19 cases during the study period, our sample size may be too small to discover a few seropositive cases. Nevertheless, we collected blood samples from people at risk only, e.g., taxi drivers, workers in entertainment venues, and airport officers, relying on the information reported by MOPH (mentioned in the Introduction). We also included health providers, workers who shared the same workplaces or communities with the SARS-CoV-2 infected cases. There was no bias for groups of participants enrolled and sample size because almost of the people concerned in an outbreak were enrolled through contact tracing performed by MOPH authorities. Furthermore, our result on the low seroprevalence of anti-SARS-CoV-2 antibodies was in line with the other two studies in Thailand. Please see our discussion in line numbers 311-322.

Lines 311-322:

Due to low prevalence of SARS-CoV-2 infection in Thailand during the study period, our sample size may be too small to discover a few seropositive cases present in a certain population. Nevertheless, our low seroprevalence data was in line with those reported by the other Thai investigators. For example, the screening for anti-SARS-CoV-2 antibodies in 6,651 royal Thai army personnel between 1 July and 30 September 2020 yielded 41 seropositive participants as determined by Wondfo® rapid diagnosis. Nevertheless, only one participant could be confirmed by Euroimmune® ELISA, resulting in a seroprevalence of 0.015% [20]. Furthermore, the serosurveillance of 600 health care providers from 4 hospitals for one year after the first case was detected in Thailand yielded only one seropositive participant (seroprevalence 0.2%) [21]. Notably, this individual was positive for IgG antibodies against the S protein but not IgG antibodies against the N protein, as determined by Euroimmune® ELISA. These seroprevalence data strongly supported the low prevalence of infection reported by the MoPH…..

2. The seroprevalence could have been due to a natural infection or vaccination. The results have to be stratified by these to clearly understand the burden of infections during the different surges.

Response: We have responded to this comment to the first reviewer. Please see line numbers 308-310.

Lines 308-310:

None of the participants had received the COVID-19 vaccine at the time of blood collection, implying that all seropositive individuals in this study were naturally infected.

Minor concerns:

1. The demographic characteristics of the participants have to be provided as table-1.

Response: Thank you for your suggestion. We include demographic characteristics of the participants in Table 1 (Line 132). 

2. It would be advisable to provide a STROBE checklist as supplementary.

Response: Thank you for your suggestion. We mention for STROBE checklist in the Experimental design of the revised manuscript (Lines 162-164) and include a STROBE checklist in Supplementary Table (S1 Table STROBE Statement).

Lines 162-164:

The findings of this study are reported according to the Strengthening the Reporting of Observational Studies in Epidemiology (STROBE) statement guidelines as shown in S1 Table.

3. A break-up of seroprevalence by the individual assays (in supplementary) will be informative to understand the discordance between the assays.

Response: Thank you for your suggestion. We add the detail of seroprevalence determined by each method in the revised manuscript. Please see “Characterization of seropositive sera” in Results, line numbers 280-283”, and supplementary Table 2 (S2 Table). 

Lines 280-283:

Characterization of seropositive sera

Of all 4964 serum samples investigated by microNT and CLIA with IFA verification, we obtained the final result of 320 seropositive samples. The seropositive results using individual assays is shown in S2 Table.

 

Reviewer #3: Reviewer’s comments on manuscript by Puthavathana and colleagues

Seroprevalence of anti-SARS coronavirus 2 antibodies in Thai adults during the first

three epidemic waves

This manuscript investigated the SARS-CoV-2 sero-prevalence in over 4,000 Thai individuals in the Bangkok and Chiang Mai regions. Various groups, including hospitalized individuals and those with high risk for exposure to infection, were sampled over the course of three epidemic waves and prior to the loss of SARS-CoV-2 containment by non-pharmaceutical interventions. Overall, the study was well executed given than three measures of sero-prevalence were used, and the manuscript is clear, but requires some editing with respect to grammatical errors.

Response: We had our manuscript edited by Springer Nature Author services. 

Minor comments:

Figure 1 – reduce the breakdown for number of days post-onset to more biologically relevant intervals. For example, 0-10 days can be group as in general, antibodies against SARS-CoV-2 are detectable 10 days post symptom onset.

Response: In our opinion, the earliest day that we can detect anti-SARS-CoV-2 antibodies is helpful for disease diagnosis when genome detection results are inconclusive. This criterion is used for non-vaccinated individuals worldwide. Our study showed that antibody testing could help diagnosis in 50% of the patients at day 7 post-infection. Therefore, we would like to keep our presentation as it was. 

I would move the description of how the results of the three serological tests were used (line 255-262) to the start of the results section.

Response: We follow your suggestion. Those sentences were moved to the paragraph on Experimental design in Material and method. Please see line numbers 152-156 in Material and method”, and Lines 286-289 in Discussion. 

Lines 152-156:

In this cross-sectional surveillance study, we employed 3 serological techniques (microNT assay, CLIA-Architect IgG, and IFA) to detect anti-SARS-CoV-2 antibodies. The neutralizing (NT) antibodies were directed against the neutralizing epitopes present in the receptor-binding domain (RBD), N-terminal domain, and S2 domain in the spike protein [9-13]; Architect IgG was directed against the nucleoprotein (N) antigen, which is a more conserved protein [14, 15].

Lines 286-289:

We employed 3 serological techniques to detect anti-SARS-CoV-2 antibodies in this cross-sectional serosurveillance study. The microNT assay and CLIA-Architect IgG were used to investigate every serum sample, and the IFA was used to verify only the sera with discordant results from the two methods.

Line 303 – please provide a reference for the statement: “We cannot deny that these outbreaks due

Response: We added 3 references as suggested. Please see line numbers 350-352 and reference numbers 31-33.

Line 350-352:

We cannot deny that these outbreaks were due to illegal activities, including the cross-border movement of migrant workers and people who gamble [31-33].

---

## [Editor Report · Decision Letter 1]

29 Mar 2022

Seroprevalence of anti-SARS-CoV-2 antibodies in Thai adults during the first three epidemic waves

PONE-D-22-01456R1

Dear Dr. Puthavathana,

We’re pleased to inform you that your manuscript has been judged scientifically suitable for publication and will be formally accepted for publication once it meets all outstanding technical requirements.

Kind regards,

Jayanta Bhattacharya

Academic Editor

PLOS ONE
---

## [Editor Report · Acceptance letter]

18 Apr 2022

PONE-D-22-01456R1 

Seroprevalence of anti-SARS-CoV-2 antibodies in Thai adults during the first three epidemic waves 

Dear Dr. Puthavathana:

I'm pleased to inform you that your manuscript has been deemed suitable for publication in PLOS ONE. Congratulations! Your manuscript is now with our production department. 

Kind regards, 

on behalf of

Dr. Jayanta Bhattacharya 

Academic Editor

PLOS ONE